# Modification of Xanthan Gum for a High-Temperature and High-Salinity Reservoir

**DOI:** 10.3390/polym13234212

**Published:** 2021-12-01

**Authors:** Mohamed Said, Bashirul Haq, Dhafer Al Shehri, Mohammad Mizanur Rahman, Nasiru Salahu Muhammed, Mohamed Mahmoud

**Affiliations:** 1Department of Petroleum Engineering, King Fahd University of Petroleum & Minerals, Dhahran 31261, Saudi Arabia; g201074520@kfupm.edu.sa (M.S.); g201907810@kfupm.edu.sa (N.S.M.); mmahmoud@kfupm.edu.sa (M.M.); 2Interdisciplinary Research Center for Advanced Materials, King Fahd University of Petroleum and Minerals, Dhahran 31261, Saudi Arabia; mrahman@kfupm.edu.sa

**Keywords:** xanthan gum, green enhanced oil recovery, polymer synthesis

## Abstract

Tertiary oil recovery, commonly known as enhanced oil recovery (EOR), is performed when secondary recovery is no longer economically viable. Polymer flooding is one of the EOR methods that improves the viscosity of injected water and boosts oil recovery. Xanthan gum is a relatively cheap biopolymer and is suitable for oil recovery at limited temperatures and salinities. This work aims to modify xanthan gum to improve its viscosity for high-temperature and high-salinity reservoirs. The xanthan gum was reacted with acrylic acid in the presence of a catalyst in order to form xanthan acrylate. The chemical structure of the xanthan acrylate was verified by FT-IR and NMR analysis. The discovery hybrid rheometer (DHR) confirmed that the viscosity of the modified xanthan gum was improved at elevated temperatures, which was reflected in the core flood experiment. Two core flooding experiments were conducted using six-inch sandstone core plugs and Arabian light crude oil. The first formulation—the xanthan gum with 3% NaCl solution—recovered 14% of the residual oil from the core. In contrast, the modified xanthan gum with 3% NaCl solution recovered about 19% of the residual oil, which was 5% higher than the original xanthan gum. The xanthan gum acrylate is therefore more effective at boosting tertiary oil recovery in the sandstone core.

## 1. Introduction

Oil production from reservoirs has been classified into three distinct phases, each determined by the use of primary, secondary, and tertiary recovery processes [1]. Primary recovery involves production driven by the in situ energy in a reservoir, with fluid flowing from the reservoir to the wellbore and thence up to the surface without any support [1]. This phenomenon is usually controlled by a natural or artificial lift system, and leads to the recovery of about 30% of the oil. With continuous field production, the secondary recovery stage is reached when the pressure in the reservoir diminishes, wherein the reservoir lacks the energy to lift the oil from the wellbore to the surface [2]. This stage involves the injection of fluid (waterflooding or gas flooding) for pressure maintenance, and it has no impact on the rock properties. At a point where the secondary process becomes uneconomical, the unrecovered oil in the form of residual oil is usually produced via a tertiary recovery process known as enhanced oil recovery (EOR) [3]. This residual oil, produced via EOR, could be categorized as isolated oil droplets, oil film, residual oil in dead-ends, residual oil in pore throats, or clusters [4,5].

The EOR technique entails the injection of chemicals not usually present in the reservoir after primary and secondary production. In contrast, green EOR (GEOR) uses environmentally friendly agents to enhance oil production [6,7,8]. However, not all reservoirs follow the exact chronological sequence in the same way as heavy oil reservoirs, where the tertiary process is usually applied as the second operation [9]. Traditional techniques, such as the use of thermal, gas, and chemical EOR, have been introduced to recover the unproduced residual oil through phase behavior change, interfacial tension (IFT) reduction, wettability alteration, and mobility modification [10]. The most common conventional chemical methods that have been employed in EOR studies by several researchers include surfactant, polymer, surfactant-polymer (SP), alkali-surfactant-polymer (ASP), and low-salinity flooding methods [1,11,12,13].

Polymer EOR techniques over the years have been the most sought after when it comes to mobility control due to their established use in field studies and their historical applications [14]. Out of the 11% of chemical EOR (cEOR) projects globally, 77% involve polymer flooding, while 23% are based on surfactant and polymer integrations [15,16]. A comprehensive review of the different polymers (synthetic and biopolymers) used for EOR applications is available in the literature [17]. However, the need for the oil industry to adopt practices that are tailored towards zero-emissions and environmental sustainability has motivated many researchers to investigate environmentally friendly polymers that can be used for the EOR process. These polymers need to be better for the environment than traditional techniques, and their use also needs to match the operational effectiveness of prior methods. Examples of such polymers include xanthan gum, schizophyllan, guar gum, scleroglucan, lignin, and welan gum, to name a few [18,19].

Nonetheless, the xanthan gum, scleroglucan, and schizophyllan classes of green polymers have a wide range of applications in the oil and gas industries due to their rigidity and stability in high-temperature environments. In addition, they possess excellent viscosity and good tolerance behavior [18]. These biopolymers also have antibacterial, antioxidant, and antitumor applications, and can be used in metal reduction as well as emulsifying and texturizing activities [20,21]. Though these recently applied groups of green polymers have exhibited excellent rheological properties in drilling and EOR studies, their application cost is considered to be the foremost barrier to operationalizing some of them, especially scleroglucan and schizophyllan. With this in mind, some authors have argued that the best choice of polymer depends on its performance and economic value [3]. Other advantages of xanthan gum and natural gum over scleroglucan and schizophyllan have been reported in various studies [19,22]. The key advantages of xanthan gum are its high viscosity, thermal stability, affordability, rheological properties, and safety, as well as its low environmental impact (see Supplemental Materials, Appendix A). This leads to the motivation for this work, where we aimed to modify xanthan gum for use in a high-salinity and high-temperature reservoir.

Herein, we present an experimental study whereby we have modified xanthan gum through a chemical process in order to scrutinize the rheological properties and EOR potential of both original xanthan gum and modified xanthan gum. The aim of this research was to develop a green polymer from xanthan gum and explore the relationship between the rheological properties and EOR performances of xanthan gum and the developed polymer. To achieve the objectives, the following steps were followed:(1)Modify xanthan gum by adding acrylic acid;(2)Examine the modified xanthan gum’s chemical structure using FT-IR and NMR spectra;(3)Investigate the rheological properties of the xanthan gum and modified xanthan gum solutions at different temperatures to determine the level at which the synthesis process has been achieved;(4)Explore the relationship between the EOR performances of the xanthan and modified xanthan gums.

## 2. Methodology

### 2.1. Materials

The NaCl was 99.87% pure and the acrylic acid was 99.99% pure, and both were purchased from Sigma Aldrich, Dammam, Saudi Arabia. The Arabian light crude oil that was used in this work was supplied by Saudi Aramco. The density of the Arab light crude was 0.87 g/cm^3^ at 23 °C and the viscosity was 19.8 mPa s at 25 °C. The dimethylaminopropyl catalyst was purchased from SOMATCHO, Dammam, Saudi Arabia. The xanthan gum that was used in the research was purchased from Sigma Aldrich, Dammam, Saudi Arabia. Gluconic acid (99.99% pure) and polyvinyl acetate (99.98% pure) were also purchased from Sigma Aldrich, Dammam, Saudi Arabia.

#### 2.1.1. Xanthan Gum

Xanthan gum (XG) is commercially produced through the process of fermentation. This process is achieved through the action of *Xanthomonas campestris* bacteria on carbohydrate substrates, such as glucose or fructose, with a protein supplement and an inorganic source of nitrogen [18,23]. The chemical structure of XG that is displayed in the Appendix A, Appendix A, shows a single glucuronic acid unit, two mannose units, and two glucose units with molar ratios of 2.0, and 2.8, respectively [17]. The thickening ability of the XG biopolymer highly depends on its molecular weight and the rigidity of the polymer chains [24]. Due to its high acceptability (non-toxic) and excellent thickening properties, it is used as a gelling agent in the food and cosmetic industries [3].

#### 2.1.2. Acrylic Acid

Acrylic acid (AA) is hygroscopic, brittle, and colorless in nature, with a boiling point of nearly 141 °C [25]. At temperatures between 200 °C and 250 °C, it loses water and becomes an insoluble, cross-linked polymer anhydride [25]. It is an industrial chemical with considerable value and huge market demand and has a wide range of applications in polymer studies. As a monomer, AA can be synthesized for use in scale inhibitors, dispersing agents, thickeners, copolymer emulsion for paints, cosmetics, papers, and so on [26,27,28,29].

#### 2.1.3. Gluconic Acid

Gluconic acid (GA) is a noncorrosive, non-toxic, mild organic acid with a translucent, brown appearance. It is produced from glucose through a simple dehydrogenation reaction, which is catalyzed by a glucose oxidase. This type of acid is very soluble in water and has a mild, refreshing taste. It is considered to be a good chelating agent, even at a high pH, compared to other chelators [30]. It has several applications especially in the food industry, where it serves as a natural constituent in the fruit juices and honey that are used in food pickling. In the petrochemical and oil and gas industry, gluconic acid, based on its salt content and derivatives, can be applied as a detergent, cement additive, and scale removing agent, and has also been used in the textile and paper industries, and many more areas. Details on its properties and application have been presented elsewhere [30].

#### 2.1.4. Polyvinyl Acetate

Polyvinyl acetate (PVA) is a synthetic, water-soluble polymer. Its backbones are composed only of carbon atoms, which are biodegradable under both aerobic and anaerobic conditions [31,32]. PVA is an odorless, non-toxic, water-soluble, and fully biodegradable polymer that has good film-forming capacity, resistance to greases and oil, and good mechanical properties, and which also acts as a good barrier to oxygen and aroma [33]. It has wide industrial applications, and is used in biopolymer film, the coating industry, food processing, the medical industries, and it has also been used as a hydrate inhibitor, thickener, and emulsion stabilizer in the oil and gas industries [31].

#### 2.1.5. N-[3-(dimethyl aminopro-165 pyl)]-N-ethyl Carbodiimide Hydrochloride

This is a water-soluble condensing reagent that is used as a carboxyl-activating agent for amide bonding with primary amines. In polymer studies, it is used as a catalyst for chemical reactions [34].

### 2.2. XG Modification

#### 2.2.1. Physical Blending

In this method, an XG solution is physically mixed with acrylic acid and gluconic acid. Xanthan gum and gluconic acid were blended at the same concentration ratio (1:1). The concentration used in the experiment was 1500 ppm. Similarly, the mixing ratios between acrylic acid and xanthan gum that were used in the experiment were 1:1, 1:3, and 1:4.

#### 2.2.2. Chemical Modification

The XG biopolymer with 701 g/mol of molar mass was balanced based on stoichiometry with AA of 56 g/mol molar mass. This chemical reaction forms a chemical bond between the carbon atom from the XG biopolymer and the oxygen atom in the AA. In addition to that, a hydrogen atom from the AA joins the hydroxyl group (OH) from the XG, forming a water molecule, as shown in Figure 1.
$$\mathrm{Acrylic}\;\mathrm{Acid}+\mathrm{Xanthan}\;\mathrm{Gum}\overset{\mathrm{Cat},70{}^{\circ}C}{\to }\mathrm{Xanthan}\;\mathrm{Acrylate}+\mathrm{Water}$$

The reaction procedure for the modification of XG using AA occurred over a number of steps (see also Appendix A). First, 1 g of xanthan gum was dissolved in 120 mL water under magnetic stirring for 22 h and heated up to a temperature of 70 °C. The XG solution was continually stirred to ensure the even distribution of the particles without suspension. A dropwise addition of AA (67 mg) was carried out before adding the N-[3-(dimethyl aminopropyl)]-N-ethyl carbodiimide hydrochloride (196 mg) catalyst to the solution, which was then stirred continuously for another 24 h. After that, 150 mL of acetone was added to the mixture and gently stirred to precipitate the polymer. Then, the solution was filtered to separate the polymer, and the product was rinsed four to five times with an ethanol–water mixture (ethanol 75 mL + water 25 mL) and then rinsed again using only ethanol (100 mL). Finally, the product was dried in a vacuum oven at 30 °C.

### 2.3. Rheological Property Measurements

The first step in investigating the changes in the chemical structure of the polymer was to study changes in the rheological properties. The discovery hybrid rheometer (DHR), as shown in the Appendix A, was an advanced combined motor and transducer rheometer consisting of a primary instrument and a separate electronics box. The DHR was used to measure the changes in viscosity that followed changes in the shear rate at different temperatures up to 80 °C, which was the limit of the equipment. After selecting cone and plate geometry, enough sample was loaded onto the cone to ensure that the rotor was lower. The air pressure was adjusted to make sure the rheometer was at zero.

### 2.4. FT-IR

The modified xanthan gum was analyzed using FTIR spectroscopy (Impact 400D, Nicolet, Madison, WI, USA). Five milligrams of the polymer were dissolved in 7 mL of deuterated chloroform. Tetrahydrofuran (THF) was used as an internal reference.

### 2.5. Core Flooding Experiment

Core flood experiments were conducted in the Core Flooding Laboratory at King Fahd University of Petroleum and Minerals (KFUPM), Saudi Arabia. The polymer flooding process is normally conducted over three stages, namely, core preparation, brine flooding, and polymer flooding. First, the core plug is cut and dried at 60 °C in an oven for four hours. Then, the dry weight is measured, and an overburden pressure is applied to the core holder. Next, the core plug is placed in a vacuum chamber to remove the air. In the brine saturation stage, a 3% NaCl solution (using deionized water, purified using the Elix 3 system, Millipore, Burlington, MA, USA) was applied to saturate each core, and, thereafter, the cores were flooded with an Arabian light crude oil until the residual water saturation of the cores were reached. Next, the oil-saturated cores were aged for three days. At this point, the core was saturated with oil and ready for flooding. The core was then flooded with 3% NaCl brine for about 2–3 pore volumes (PV). Typically, the brine flooding was stopped when only a trace amount of oil was being produced. In the last stage, the polymer formulation (xanthan gum or xanthan acrylate with 3% NaCl) was used to flood the core for about 2–3 PV at an injection rate of 0.50 cm^3^/min. Post-flooding with brine was conducted afterward to ensure that no oil remains in the tube and core and gets collected in the accumulator. Figure 1 outlines the core flood process and Figure 2 describes the experimental setup.

## 3. Results and Discussion

### 3.1. FT-IR and NMR Analysis

The Fourier transform-infrared spectrum (FT-IR) is a well-known method for detecting similarities and/or differences in compound chemical structures. In this section, the FT-IR analysis performed on both the xanthan and modified xanthan gums to investigate the chemical modification is described. The FT-IR spectra of acrylic acid, xanthan gum, and modified xanthan gum are shown in Figure 3. Acrylic acid has a band at 1698 cm^−1^, which represents the C=O stretch, followed by bands at 1635, 1239, 1048, 981, 925, 813, and 648 cm^−1^, which correspond to the C=C stretch, CH in-plane bend, CH_2_ rocking, out-of-phase CH_2_ wag, out-of-plane CH bend, out-of-plane OH bend, CH_2_ twist, and CO_2_ in-plane bend, respectively. The FT-IR spectrum for the xanthan gum (see Figure 3) shows absorption peaks at 3277 cm^−1^, which are due to the O–H axial deformation; a similar phenomenon at 2850–2950 cm^−1^ is due to the symmetric and asymmetric stretching vibrations of the C–H group in the methyl and methylene groups. The bands at 1710 cm^−1^ are due to the C=O stretching vibrations, while bands nearer to 1601 cm^−1^ are due to the axial deformation of the C–O part of the enols. The FT-IR spectrum for the modified xanthan gum, as displayed by Figure 3, shows peaks identical to those of the xanthan gum. Additionally, a new peak appeared at 1645 cm^−1^, corresponding to the C=C, which came from the acrylic acid. This confirmed a successful modification of the xanthan gum with the acrylic acid. Figure 3 shows a clear distinction between the xanthan gum and the modified xanthan gum polymers.

### 3.2. Rheological Property

Xanthan gum viscosity is considered to be the main factor for evaluating the effectiveness of a polymer flood, since the primary goal of adding polymers to water is to increase its viscosity and thus decrease the mobility ratio [3]. The result is that the water sweep efficiency rises. Temperature is one of the key parameters that influence polymer viscosity. The temperature effects on the physical and chemical modifications of xanthan gum are demonstrated thoroughly in the following sections.

#### 3.2.1. Blending of XG with PVA, GA, and AA

The change in viscosity of the xanthan gum was investigated by varying the temperature at different shear rates, as shown in the Appendix A. The temperature was adjusted to 25 °C, 40 °C, 50 °C, 60 °C, 70 °C, and 80 °C. For each test, 25 mL of xanthan solution was used. It was observed that as the temperature increased, the solution’s viscosity decreased. For example, at a shear rate of 1 s^−1^, the solution’s viscosity was measured as 5 cp and 1.8 cp at 25 °C and 80 °C, respectively. It was found that the difference in viscosity with the shear rate from 0.1 s^−1^ to 1000 s^−1^ at a given temperature was power-related.

To study the effect of salinity at a given temperature, NaCl was added to the 1% total weight of the xanthan gum solution, and the viscosities were measured at different shear rates. From the experiments, it was determined that up to moderate temperatures (i.e., 50 °C) and when applying lower shear rates, adding NaCl had a noticeably negative impact on the viscosity of the gum, as shown in the Appendix A, Appendix A. However, as the temperature increased, the effect of the NaCl was minimized, and the viscosity appeared to be almost the same as when no NaCl was added, as shown in the Appendix A, Appendix A. For example, at 50 °C and with a 0.01 s^−1^ shear rate, the gum viscosity was 48.34 mPa·s, while the viscosity of the gum with the NaCl under the same conditions was 36.12 mPa·s. The Newtonian viscosity region was observed at a low shear rate (i.e., 0.04 s^−1^). Moreover, as the temperature increased, the effect of the salt solution decreased until it was almost negligible at 80 °C, as shown in the Appendix A, Appendix A.

The temperature was kept constant to study the effect of adding PVA on the viscosity of XG at varying shear rates. It was found that the blending effect of PVA in the XG solution was minimal because the viscosity of the PVA and XG blend did not increase significantly compared to pure XG. For example, at 50 °C and with a 1000 s^−1^ shear rate, the xanthan gum’s viscosity was 0.018 mPa·s, while for the xanthan gum with PVA added, the viscosity was 0.020 mPa·s, as shown in the Appendix A, Appendix A. In addition, at 80 °C and with a 0.01 s^−1^ shear rate, the xanthan gum’s viscosity was 10.41 mPa·s, while the xanthan gum with PVA had a viscosity of 11.37 mPa·s, as shown in the Appendix A, Appendix A. However, the change in viscosity was minimal and thus considered to be negligible.

The viscosity of the XG was inversely proportional to the temperature and shear rate. Moreover, the experimental results from the effect of NaCl on the viscosity of the solution, which simulated the injected water, show that the addition of a 3% salt solution slightly decreased the viscosity of the xanthan gum solution. Furthermore, the results show that adding PVA barely increased the polymer’s viscosity, to the point that the change was negligible.

GA has a carboxyl group in its chemical structure, which was the reason why it was chosen. It was expected to connect to the xanthan gum’s chain and increase the strength of the bonds between the molecules, and the viscosity was expected to increase as a result. This might happen with a chemical reaction. However, blending the xanthan gum with gluconic acid at the same concentration ratio (i.e., 1:1) slightly decreased the viscosity of the gum, as highlighted in the Appendix A, Appendix A. The concentration used in the experiment was 1500 ppm (0.15%).

It was observed that mixing XG with AA did not increase the viscosity of the solution. Rather, it decreased the viscosity as the concentration of AA increased. The xanthan gum concentration used in the experiments was 1500 ppm and the temperature was 25 °C. The volume ratios of xanthan gum to AA used in the experiments were 1:1, 1:3, and 1:4, as shown in the Appendix A. As the concentration of AA was increased in the blending process, the viscosity of the solution decreased. This reduction in viscosity that occurred after adding the AA was because of the reaction between the xanthan gum and the AA. Therefore, the mixed solution became more diluted (1:4) and thus the viscosity decreased. A similar trend was observed at 50 °C, as shown in the Appendix A.

#### 3.2.2. Chemically Modified Xanthan Gum—Xanthan Acrylate

The xanthan gum was modified with acrylic acid in a chemical reaction, as described earlier. The chemically modified xanthan gum is called xanthan acrylate and the results of the analysis are discussed below.

The viscosities of the xanthan acrylate and xanthan gum were measured at 25 °C and 50 °C and the concentration of both polymers was kept constant at 1500 ppm. Figures showing the viscosity of both at 25 °C and 50 °C can be found in the Appendix A. The xanthan acrylate had a higher viscosity value than the xanthan gum at the same shear rate. This improvement in viscosity was due to a chemical reaction between the xanthan gum and the acrylic acid that connected the hydroxyl group to the xanthan gum’s chain and made them stronger. Therefore, the viscosity increased.

The modified polymer offered higher stability, in terms of viscosity, with the temperature change, as shown in Figure 4. For example, at a shear rate of 0.1 s^−1^, the viscosities of the xanthan gum were 578 and 205 mPa·s at 25 °C and 50 °C, respectively. The xanthan acrylate’s viscosities were 2200 and 1070 mPa s at 25 °C and 50 °C, respectively. The reduction in xanthan gum’s viscosity was around 65%. For the xanthan acrylate, the decrease in viscosity was around 50%. Even at a 10 s^−1^ shear rate, the xanthan gum’s viscosities were 63.5 and 44 mPa·s at 25 °C and 50 °C, respectively. For the xanthan acrylate, they were 182 and 130 mPa·s at 25 °C and 50 °C, respectively. The decrease in xanthan gum’s viscosity was around 30%. For the xanthan acrylate, the decrease in viscosity was around 20%.

It is evident from the rheological experiments into the physical and chemical modifications that only the latter succeeded in increasing the xanthan gum’s viscosity. Consequently, a comparison between the xanthan gum and xanthan acrylate was conducted using core flood experiments to investigate which two led to better oil recovery.

### 3.3. Core Flooding Results

Core flooding simulates reservoir conditions and therefore helps investigate the EOR performances of polymer systems. Two core flood experiments were conducted to analyze and compare the recoveries of the pure xanthan and modified xanthan gums. The following sections present the results of the experiment into the recoveries of xanthan acrylate and pure xanthan gum and compare their EOR performances.

#### 3.3.1. Oil Recovery of the Xanthan Acrylate

First, the core was saturated with oil and 3% NaCl brine and then flooded by three pore volumes (PV) of 3% NaCl until no oil was produced. Then the core was flushed with 3 PV of xanthan acrylate until no more oil was produced. After that, the core was flooded by 1 PV of 3% NaCl brine to ensure that all of the mobile oil had been produced.

The recovery obtained via water flooding was 43.85% of the initial oil in place (IOIP), and the tertiary oil recovery due to the xanthan acrylate flood was 19.21% of the IOIP. The total oil recovery by the water and xanthan acrylate was 63.1% of the IOIP. Figure 5 shows the recoveries in percentages, and Table 1 summarizes the core flood results of the xanthan acrylate solution experiment.

#### 3.3.2. Comparison of Pure Xanthan Gum and Xanthan Acrylate Recoveries

The oil recovery performance of the modified xanthan gum was compared with the pure xanthan gum. A core flooding experiment was conducted using 3% NaCl and 1500 ppm pure xanthan gum solution. It was found that at the end of the waterflood, 44.88% of the IOIP was recovered, and at the end of the pure xanthan gum solution flood, the tertiary oil recovery was 13.93%. The total recovery of this formulation was 58.81% of the IOIP. Figure 6, alongside Appendix A in the Appendix A, demonstrates the recovery performance of pure xanthan and xanthan acrylate solutions. It is found from the literature that oil recovery by hydrolyzed polyacrylamide (HPAM) is usually around 12% to 17% [35].

The core flood results for the two formulations are given in Table 1 and Figure 6. It is clear from Figure 6 that the xanthan acrylate and 3% NaCl blend produced 19.21% of the IOIP as tertiary oil. However, the pure xanthan gum and 3% NaCl solution recovered 13.93% of the IOIP as tertiary oil, which was 5.28% less than the xanthan acrylate solution. Therefore, by comparing the two formulations, it can be concluded that the xanthan acrylate and 3% NaCl formulation were more effective at recovering tertiary oil from the sandstone core.

## 4. Conclusions

This study experimentally investigated the prospects for the use of xanthan gum modification in a high-salinity and high-temperature reservoir. Based on this research, the main findings of this paper are:(1)The FT-IR and NMR analyses showed that there was a successful chemical modification of the xanthan gum. The chemical structure of the xanthan gum was reformed after its reaction with acrylic acid. The FT-IR spectrum of the xanthan gum showed absorption peaks at 3277 cm^−1^. The FT-IR spectrum of the xanthan acrylate showed peaks identical to those of the xanthan gum. Moreover, a new peak appeared at 1645 cm^−1^.(2)The chemically modified xanthan gum demonstrated increased rheological properties, compared to the xanthan gum. The viscosity at a 0.1 s^−1^ shear rate and 25 °C was 578 mPa·s for the xanthan and 2200 mPa·s for the xanthan acrylate under the same experimental conditions. In addition, the stability in terms of viscosity of the xanthan gum increased with increasing temperatures.(3)The improvement of the xanthan gum’s viscosity after its modification was reflected in the oil recovered from the cores. The characteristics of the rocks were kept the same to allow for a reasonable comparison between the original and modified xanthan gums. The core flooding results also confirmed that the modified xanthan gum offered a higher rate of oil recovery, recovering more than 5% more of the IOIP compared to the pure xanthan gum due to the increased viscosity. The xanthan gum in combination with the 3% NaCl solution recovered about 45% of the oil as secondary oil and 14% as tertiary oil. The modified xanthan gum formulation produced about 44% of the oil as secondary oil and 19% as tertiary oil.

## 5. Recommendations

(1)The proposed modification for xanthan gum should be tested on carbonate rock.(2)Further research should be conducted to identify other modified xanthan gums with increased concentrations of acrylic acid.(3)After many trials, the chemical reaction was successful because it was susceptible to temperature. Therefore, future work should take into consideration adjustments of the reaction temperature.

## Data Availability

Data sharing not applicable.

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
