# Peer review of "Modification of Xanthan Gum for a High-Temperature and High-Salinity Reservoir"

_polymers, 2021, doi:10.3390/polym13234212_

Round 1

Reviewer 1 Report

Comments:

1. The average molecular weight of the modified xanthan gum should be compared with the pristine gum with GPC (gel permeation chromatography) method.

2. The manuscript should be checked for typo errors

Author Response

RESPONSE TO THE REVIEWER COMMENTS – Round 3

REVIEWER 1

Manuscript ID: polymers - 1333539

First, we would like to thank the reviewer for his/her favourable evaluation of our work and valuable suggestions. In the revised manuscript, we have addressed the specific comment made by the reviewer. In response to the reviewer comments, modified text in the manuscript is marked in red and heightened in yellow.

COMMENT 1

Introduction: Please add few sentences for the importance of natural gums, including xanthan gum, for the enhanced oil recovery.

Response:

A number of sentences (Lines 73-74) are added to discuss the advantage of Xanthan gum and natural gum in the introduction.

COMMENT 2

General comment: the manuscript has language and typo errors.

Response:

Please accept our sincere apology. The revised manuscript is edited by MDPI English Editing. MDPI takes responsibility for the editing carried out on the paper.

COMMENT 3

Provide the average molecular weight of chemically modified xanthan (xanthan gum acrylate) by the GPC (gel permeation chromatography) method.

Response:

Please accept our sincere apology. The student is already graduated, and he is no longer available to work on it. It is difficult to provide the information at this stage. However, we are searching for other options, and if we can get it, we will add it in the text.

COMMENT 4

Research highlights should be removed from the abstract section.

Response:

The highlight is deleted from the abstract.

COMMENT 5

Revise the Fig. 1. The text font should be adjusted to make the legend more precise.

Response:

Fig 1 is revised, and the text font is adjusted.

COMMENT 6

Please provide a composite Figure (FT-IR analysis) for Fig. 5 to Fig. 8.

Response:

According to the suggestion, a composite figure (Now referred to as Figure 3) for FT-IR analysis is added in the text.

COMMENT 7

Table 1. EOR by xanthan gum, and Table 2. EOR by NP should be combined to make a single Table for EOR comparison.

Response:

Tables 1 and 2 are combined into one table and added in the main text.

COMMENT 6

What do you mean by NP (new polymer)? It would be nice to use ‘chemically modified xanthan gum’ instead of NP.

Response:

 The new polymer is called Xanthan Acrylate, which is chemically modified xanthan gum. The Xanthan Acrylate is used instead of NP in the main text to make it short and sweet.

Reviewer 2 Report

General remarks

Dear authors,

From the marked parts in the text I assume this is a revised version of the manuscript. However, in my opinion some more careful revision is needed before publication. Generally I think the topic is interesting, especially as it aims at practical use (as oil still is the main provider for mobility) and thus the results of your experiments should be made available for a wider audience.

At places I had some problems to understand your intentions (s. Specific Remarks), which apart from some inaccuracies, result from some problems with use of English language. I made some suggestions (s. Specific Remarks), but I would recommend to have a native speaker or someone fluent in English reading the manuscript.

The Figures are sufficient, however, two times I would recommend to combine two single figures to make it more easy to see the results and to compare them (s. Specific Remarks).

Finally, just as a general comment. A major topic is the viscosity. Only, all over the manuscript you wrote “mPa. S” instead of “mPa s”! Please be more careful!

Specific Remarks

For my Specific Remarks, please see the attached pdf document I added my comments to!

Author Response

RESPONSE TO THE REVIEWER COMMENTS – Round 3

REVIEWER 2

Manuscript ID: polymers-1333539

First, we would like to thank the reviewer for his/her favorable evaluation of our work and valuable suggestions. In the revised manuscript, we have addressed the specific comment made by the reviewer. In response to the reviewer comments, modified text in the manuscript is marked in red.

COMMENT 1

my Specific Remarks, please see the attached pdf document I added my comments to!

Response:

COMMENT 3

just as a general comment. A major topic is the viscosity. Only, all over the manuscript you wrote “mPa. S” instead of “mPa s”! Please be more careful

Response:

This has been carefully addressed in the manuscript.

COMMENT 4

I would recommend to have a native speaker or someone fluent in English reading the manuscript.

Response:

The MDPI English Editing edits the manuscript. The name of the English Editor is Peter Russel.  He is working in the MDPI English Editing. Peter has edited the manuscript and the English editing reference no is 33531.

COMMENT 5

Line 36: I would suggest to write "...and has no impact on rock properties"

Response:

Thank you for the observation. The suggestion has been included in the manuscript. Line 36

COMMENT 6

Line 46: Please check this formulation, the sentence seems to be somewhat incomplete

Response:

Thank you for the vital observation. The term “EOR” have been added in Line 47

COMMENT 7

Line 58: Please check this “sustenability”

Response:

This has been addressed. Line 59

COMMENT 8

Line 74: I am sorry, but what do you mean by "good pH"? Please check!

Response:

The particular property is actually for Guar gum and not Xanthan.

Thank you for this observation, this have been removed both in the manuscript (Line 75) and supplementary Figure S1.

COMMENT 9

Line 84: I think I know what you mean, however, the sentence as it is now is not correct, xanthan cannot be synthesized by adding AA! Please check!

Response:

This has been addressed, the term “modified” have been adopted all through the manuscript. (Line 85).

COMMENT 10

Line 97: Please use the same unit - mPa s -  throughout the text!

Response:

This have been addressed (Line 97) and in the entire manuscript as well.

COMMENT 11

Line 145: Xnathan itself is a polymer! Do you mean modification of Xnathan? Please check!

Response:

Yes, this has been addressed (Line 145)

COMMENT 12

Line 120: GA is not listed above!

Response:

A concise section is presented for GA (section 2.1.3.) in the methodology.

COMMENT 13

Paragraph 2.3. Please check the format here!

Response:

The entire paragraph has been formatted to match MDPI polymer standard.

COMMENT 14

Line 173: Please give more information about the instrument used!

Response:

Thanks for this comment. However, the authors feel DHR (Line 173) is a conventional rheology equipment which requires no in-dept information.

COMMENT 15

Line 178: The last sentence of this paragraph can be deleted!

Response:

Thanks for the observation. The last sentence (Finally, measurement is started by clicking the START button, and the results are displayed in the 175 form of graph and the data are saved in a file) has been deleted from the revised manuscript.

COMMENT 16

Line 181: Fourier Transform (capital letter)

Response:

The capital letter is noted (Line 181)

COMMENT 17

Paragraph 2.4: Please check font size for this paragraph!

Response:

The entire paragraph has been formatted to match MDPI polymer standard.

COMMENT 18

Line 187-198: Please check this paragraph, the procedure described is not fully clear for me!

Response:

The paragraph has been described for clarity.

Core flood experiments were conducted in the Core Flooding Laboratory at King Fahd University of Petroleum and Minerals (KFUPM), Saudi Arabia. The polymer flooding process is normally conducted in three stages. These are core preparation, brine flooding, and polymer flooding. First, the core plug is cut and dried at 60 oC in an oven for four hours. Then, the dry weight is measured, and an overburden pressure is applied to the core holder. Next, the core plug is placed in a vacuum chamber to remove the air. In the brine saturation stage, 3% NaCl solution (using deionized water, purified by Elix 3 system, Millipore, USA) was applied to saturate each core, and thereafter, the cores were flooded with an Arabian light crude oil until the residual water saturation of the cores were reached. Next, the oil-saturated cores were aged for three days. At this point, the core was saturated with oil and is ready for flooding. The core is then flooded with 3% NaCl brine for about 2 - 3 pore volume (PV). Typically, the brine flooding was stopped when only a trace amount of oil was being produced. In the last stage, the polymer formulation (Xanthan gum or Xanthan Acrylate with 3% NaCl) was used to flood the core for about 2-3 PV at an injection rate of 0.50 cm3/min. Post-flooding with brine was done afterward, to ensure no oil remains in the tube and core and is collected in the accumulator. Figure 1 demonstrates the core flood process and Figure 2 describes the experimental setup.

Figure 1. Process flow diagram of the polymer flooding experiment.

Figure 2. Core flood experimental setup.

COMMENT 19

Fig 1: i. Please check for use of capital letters! ii. cm3? iii. Perhaps you better should use Pa as unit for pressure

Response:

Thanks for the observation. The units have been addressed in Figure 1, and Figure 2 is added for clarity.

COMMENT 20

Fig 3: The numbers are missing at the y axis of the left figure.

Response:

The number of Y-axis [Transmittance (%)]is given in the main plot compare the transmittance between pure xanthan and modified xanthan gums.

COMMENT 21

Line 237: This can be deleted “ A few”

Response:

This has been addressed in Line 237.

COMMENT 22

Fig 3: I would recommend to combine Figure 3 and Figure 4 and show both NMR spectra in a common graph. This would make it much more easy to see differences!

Response:

Both Figures have been combined for ease and clarity as suggested and now represent Figure 4.

COMMENT 23

Line 268: Please use at least once the full term!

Thank you for this key observation. This has been addressed in the entire revised manuscript as suggested. Also, a separate section (section 2.1.4) for PVA have been included in the methodology.

Response:

COMMENT 24

Line 246: I am sorry, but why not simply write "°C" ??

Response:

The temperature unit all through the manuscript as well as the supplementary Figures have been corrected.

COMMENT 25

Line 256. Do you mean NaCl solution? (Please check other places for this as well!)

Response

Thank you for this key observation. Yes, and this has been checked all through the manuscript.

COMMENT 26

Line 260: I would suggest to mention what kind of effect, especially as here a figure in the Supplementary is referred to!

Response:

The effect is the viscosity, (Line 261) and it was mentioned in the section.

COMMENT 27

Line 265 I would think it should be "salt solution"

Response:

Yes, this has been changed to salt solution (Line 266)

COMMENT 28

Line 271: compared to the viscosity of...

Response:

This has been revised (Line 271).

COMMENT 29

Line 278-279: This sentence rather sounds like Material & Methods, I would suggest to reformulate it a bit!

Response:

The sentences has been revised as shown below. (Lines 278 - 280)

Moreover, the experimental results from the effect of NaCl on the viscosity of the solution to simulate the injected water show that, the addition of 3% salt solution slightly decreased the viscosity of the xanthan solution.

COMMENT 30

Line 288: I would suggest to write, e.g., "... which was the reason that it was chosen" . Do you mean the xanthan chain? Please check!

Response:

The suggestion has been implemented. (Line 283)

COMMENT 31

Line 302-303: I am sorry, but whyt precisely do you mean by "non-reaction"? Especially as AA was chosen for its caboxyl group that binds with the xanthan chain and the next paragraph that states a reaction between xanthan and AA. (s. above). You here should give the amount of dilution!

Response:

Thank you very much for this careful observation. The section is revised as shown below.

As the concentration of AA was increased in the blending process, the viscosity of the solution decreased. This reduction in viscosity resulting from adding the AA resulted from the reaction between the xanthan and AA. Therefore, the mixed solution became more diluted (1:4) and thus the viscosity decreased. A similar trend was observed at 50 °C as shown in Figure S14.

COMMENT 32

Line 313: Do you mean "viscosity"? Please check!

Response:

Yes. See (Line 305)

COMMENT 33

Line 316: Do you mean the xanthan chain? Please check!

Response:

Yes, see (Line 308)

COMMENT 34

Line 336: Is there a citation for this?

Response:

Unfortunately, no reference. However, the authors believe this is a general understanding. (Line 328)

COMMENT 35

Line 349: This term only is explained in the Abstract! Please give the full term here as well!

Response:

The meaning of HPAM has been included. (Line 354)

COMMENT 36

Line 356: I would suggest to write "pure xanthan gum and ..."

Response:

It has been changed to pure xanthan gum and all other sections in the manuscript (Line 349).

COMMENT 37

Line 366: I am sorry, but what is chemical flooding? Please check!

Response:

Thank you very much. It has been addressed underneath Table 1 in the manuscript. (Line 366).

COMMENT 38

Line 379: I would think it should rather be "this study"

Response:

Thank you very much. It has been revised. (Line 368)

COMMENT 39

Line 379: I would suggest to write, e.g., ... the effect of..."

Response:

Thank you very much. The suggestion has been implemented. (Line 368)

COMMENT 40

Line 384: Please check this term!

Response:

Thank you very much for the observation. The term “connected” has been revised to present a better understanding of the statement. (Line 373)

COMMENT 41

Line 388: I am sorry, but I do not understand this! “mirroring that found for the xanthan gum”.

Response:

Thank you very much for the observation. The statement has been revised for better understanding. (Line 377)

COMMENT 42

Line 393: Stability in what respect? Please check!

Response:

Thank you very much for the observation. The Last sentence has been revised as shown below (Line 377)

In addition, the stability in terms of viscosity of the xanthan gum increased with increasing temperature.

COMMENT 43

Line 399: by more

Response:

Thank you very much for the observation. The word has been revised (Line 388)

COMMENT 44

Line 405: Which reaction?

Response:

The sentence has been edited. (Line 394)

COMMENT 45

Line 408: “to”

Response:

Thank you very much for the observation. The “to” has been changed to “of” (Line 397)

Reviewer 3 Report

This manuscript discusses about modification of Xanthan Gum for a high temperature and salinity reservoir. Manuscript needs following changes.

  1. Abstract: Improve this section by mentioning key results only.
  2. Line 11. Rewrite it as: Tertiary oil recovery, commonly known as enhanced oil recovery (EOR), is introduced when secondary recovery is no longer economically viable and water or gas is injected.
  3. Line 36: Primary recovery involves production resulting from…add reference in support …Exploiting Microbes in the Petroleum Field: Analyzing the Credibility of Microbial Enhanced Oil Recovery (MEOR)
  4. Introduction: discuss about other applications of exopolysaccharides in brief.
    Exopolysaccharide from psychrotrophic Arctic glacier soil bacterium Flavobacterium sp. ASB 3-3 and its potential applications. Bioprospecting of exopolysaccharide from marine Sphingobium yanoikuyae BBL01: production, characterization, and metal chelation activity.
  5. Line 115: Xanthomonas campestris…write in italics
  6. There is need of small subsections for each chemicals 2.1.1-2.1.5. Cobine all information under a section 2.1 Materials and mention chemical name and company name only. There is no need to add information general information about properties of chemicals. Remove general information related to acrylic acid.
  7. Section 2.2.1: Add more information as in what ratio they were mixed.
  8. Line 151-165: Write it in a single paragraph and remove numbering. Information should be in past tense.
  9. Section 2.3: Write information in past tense.
  10. Combine section 3.1 and 3.1.1 and add more discussion with references in support.
  11. Add discussion point in all sections of 3.2 with references in support.

   12. Figure 6: Remove grid line from figure 6, 7, 8

  1. Section 3.3: Add more discussion points and compare your results with other reports.

Author Response

RESPONSE TO THE REVIEWER COMMENTS – Round 3

REVIEWER 3

Manuscript ID: polymers-1333539

First, we would like to thank the reviewer for his/her favorable evaluation of our work and valuable suggestions. In the revised manuscript, we have addressed the specific comment made by the reviewer. In response to the reviewer comments, modified text in the manuscript is marked in black and heightened in green.

COMMENT 1

Abstract: Improve this section by mentioning key results only.

Response: The abstract below is edited meticulously and added in the manuscript.

Tertiary oil recovery, commonly known as enhanced oil recovery (EOR), is introduced when secondary recovery is no longer economically viable. Polymer flooding is one of the EOR methods that improve the injected water's viscosity and boost oil recovery. Xanthan gum is a relatively cheap biopolymer and is suitable for oil recovery at limited temperature and salinity. This work aims to modify xanthan gum to improve its viscosity for high temperature and salinity reservoirs. The xanthan gum was reacted with acrylic acid in the presence of a catalyst and formed xanthan acrylate. The chemical structure of the xanthan acrylate was verified by FT-IR and NMR analysis. The discovery hybrid rheometer (DHR) confirmed that the viscosity of the modified xanthan gum was improved at elevated temperature, which was reflected in the core flood experiment. Two core flooding experiments were conducted using 6-inch sandstone core plugs and Arabian light crude oil. The first formulation, the xanthan gum with 3% NaCl solution, recovered 14% residual oil from the core. In contrast, the modified xanthan gum with 3% NaCl blending produced about 19% residual oil, which was 5% higher than the original Xanthan gum. The xanthan gum acrylate is efficient in boosting tertiary oil in the sandstone core.

COMMENT 2

Line 11. Rewrite it as: Tertiary oil recovery, commonly known as enhanced oil recovery (EOR), is introduced when secondary recovery is no longer economically viable and water or gas is injected.

Response:

The sentence in Line 11 is rewritten as:

Tertiary oil recovery, commonly known as enhanced oil recovery (EOR), is introduced when secondary recovery is no longer economically viable.

COMMENT 3

Line 36: Primary recovery involves production resulting from…add reference in support …Exploiting Microbes in the Petroleum Field: Analyzing the Credibility of Microbial Enhanced Oil Recovery (MEOR)

Response:

Reference is added in the Line 31.

COMMENT 4

Introduction: discuss about other applications of exopolysaccharides in brief. Exopolysaccharide from psychrotrophic Arctic glacier soil bacterium Flavobacterium sp. ASB 3-3 and its potential applications. Bioprospecting of exopolysaccharide from marine Sphingobium yanoikuyae BBL01: production, characterization, and metal chelation activity.

Response:

This sentence is added in Lines 67 to 69.

These biopolymers also have applications for antibacterial, antioxidant, antitumor, metal reduction as well as emulsifying and texturizing activity [18,19].

COMMENT 5

Line 115: Xanthomonas campestris…write in italics

Response:

The Xanthomonas campestris is written in italic in line 104 and highlighted in light blue.

COMMENT 6

There is need of small subsections for each chemicals 2.1.1-2.1.5. Combine all information under a section 2.1 Materials and mention chemical name and company name only. There is no need to add information general information about properties of chemicals. Remove general information related to acrylic acid.

Response:

The sections from 2.1.1 to 2.1.5 are combined to a section 2.1 Materials. The general information of the chemicals is removed except xanthan gum and the section is revised according to the suggestion below and added in the manuscript.

2.1. Materials

The NaCl was 99.87% pure and acrylic acid was 99.99% pure, and both were purchased from Sigma Aldrich, Dammam, Saudi Arabia. The Arabian light crude oil was used in this work and supplied by Saudi Aramco. The density of the Arab light crude was 0.87 g/cm3 at 23 oC and viscosity was 19.8 mPa s at 25 oC. The Dimethylaminopropyl catalyst was purchased from SOMATCHO, Dammam, Saudi Arabia. The xanthan gum was used in the research and was purchased from Sigma Aldrich, Dammam, Saudi Arabia. Gluconic acid (99.99% pure) and Polyvinyl acetate (99.98% pure) were also purchased from Sigma Aldrich, Dammam, Saudi Arabia.

2.1.1. Xanthan Gum

Xanthan gum (XG) is commercially produced through the process of fermentation. This process is achieved through the action of Xanthomonas campestris bacteria on carbohydrate substrates such as glucose or fructose with a protein supplement and an inorganic source of nitrogen [18,23]. The chemical structure of XG displayed in Figure S2 shows a single glucuronic acid unit, two mannose units, and two glucose units of molar ratio 2.0, 2.0, and 2.8, respectively [17]. The thickening ability of the XG biopolymer highly depends on its molecular weight and the rigidity of the polymer chains [24]. Due to their high acceptability (non-toxic) and excellent thickening properties, they are used as gelling agents in the food and cosmetic industries [3].

2.1.2. Acrylic Acid

Acrylic acid (AA) is hygroscopic, brittle, and colorless in nature, with a boiling point of nearly 141 °C [25]. At temperatures between 200 °C and 250 °C, it loses water and becomes an insoluble cross-linked polymer anhydride [25]. It is an industrial chemical with considerable value and huge market demand and has a wide range of applications in polymer studies. Being a monomer, AA can be synthesized for use as scale inhibitors, dispersing agents, thickeners, copolymer emulsion for paints, cosmetics, papers, and so on [26–29].

2.1.3. Gluconic Acid

Gluconic acid (GA) is a noncorrosive, nontoxic, mild organic acid with a brown clear appearance. It is produced from glucose by a simple dehydrogenation reaction catalyzed by a glucose oxidase. This type of acid is very soluble in water and has a mild refreshing taste. It is considered a good chelating agent even at high pH as compared to other chelators [30]. It has received several applications especially in the food industry where it serves as a natural constituent in fruit juices and honey used in food pickling. In the petrochemical and oil and gas industry, Gluconic acid based on its salt content and derivatives can be applied as detergent, cement additives, scale removing agent, textile, and paper industry, and many more areas. Details on its properties and application are presented elsewhere [30].

2.1.4. Polyvinyl Acetate

Polyvinyl acetate (PVA) is a synthetic water-soluble polymer. Its backbones are composed only of carbon atoms, which are biodegradable under both aerobic and anaerobic conditions [31,32]. PVA is an odorless, non-toxic, water-soluble, and fully biodegradable polymer that presents good film-forming capacity, resistance to greases and oil, good mechanical properties, and a good barrier to oxygen and aroma [33]. It has received wide industrial applications such as biopolymer film, coating industry, food processing, medical industries, and it has also been used as hydrate inhibitor, thickener, and emulsion stabilizers in oil and gas industries [31].

2.1.5. Dimethylaminopropyl

This is a water-soluble condensing reagent used as a carboxyl activating agent for amide bonding with primary amines. In polymer studies, it is used as a catalyst for chemical reactions [34].

COMMENT 7

Section 2.2.1: Add more information as in what ratio they were mixed.

Response: The following mixing information is added in section 2.1.1.

The blending of xanthan gum with Gluconic acid was the same concentration ratio (1:1). The concentration used in the experiment was 1,500 ppm. Similarly, the mixing ratios between acrylic acid and xanthan used in the experiment were 1:1, 1:3, 1:4.

COMMENT 8

Line 151-165: Write it in a single paragraph and remove numbering. Information should be in past tense.

Response:

The lines 158-168 are written is a paragraph and added in the text.

The reaction procedure (Figure S3) for the modification of XG using AA is described in a number of steps. First, 1 g of xanthan gum in 120 mL water was dissolved under magnetic stirring for 22 h and heated up to a temperature of 70 °C while stirring the XG solution to ensure the even distribution of the particles without suspension. A dropwise addition of AA (67 mg) was carried out before adding N-[3-(dimethyl aminopropyl)]-N-ethyl carbodiimide hydrochloride (196 mg) catalyst to the solution and stirred continuously for another 24 h. After that, 150 ml of acetone was added to the mixture and gently stirred to precipitate the polymer. Then, the solution was filtered to separate the polymer and rinsed the product alternatively 4/5 times with ethanol/water mixture (ethanol 75 ml + water 25 ml) and then rinsed the product again using only ethanol (100 ml). Finally, the product was dried in a vacuum oven at 30 °C.

COMMENT 9

Section 2.3: Write information in past tense.

Response:

The section 2.3 is rewritten in past tense and highlighted in blue according to the suggestion. Thanks.

COMMENT 10

Combine section 3.1 and 3.1.1 and add more discussion with references in support.

Response:

The sections 3.1.1 and 3.1.2 are combined into section 3.1 FT-IR and NMR Analysis, and discussed more detail.

Lines 212 to 241 contains a detail description with reference.

COMMENT 11

Add discussion point in all sections of 3.2 with references in support.

Response:

All sections of 3.2 are revised and discussed meticulously. Lines 242 – 326.

3.2. Rheological Property

Xanthan viscosity is considered the main factor in evaluating the effectiveness of a polymer flood since the primary goal of adding polymers to water is to increase its viscosity and thus decrease the mobility ratio [3]. The result is that the water sweep efficiency rises. Temperature is one of the key parameters that influence polymer viscosity. Temperature effects on physical and chemical modifications of xanthan gum are demonstrated thoroughly in the following sections.

3.2.1. Blending of XG with PVA, GA, and AA

The change in viscosity of the xanthan gum was investigated by varying the temperature at different shear rates, as shown in Figure S5. The temperature was adjusted to 25 °C, 40 °C, 50 °C, 60 °C, 70 °C, and 80 °C. For each test, 25 ml of xanthan solution was used. It was observed that as the temperature increased, the solution viscosity decreased. For example, at a shear rate of 1 s-1, the solution viscosity was measured as 5 cp and 1.8 cp at 25 °C and 80 °C, respectively. It was found that the difference in viscosity with the shear rate from 0.1 s-1 to 1000 s-1 at a given temperature was power-related.

To study the effect of salinity at a given temperature, NaCl was added to the 1% total weight of xanthan gum, and the viscosities were measured at different shear rates. From the experiments, it was determined that up to moderate temperatures (i.e., 50 °C) and at lower applied shear rates, adding NaCl had a noticeably negative impact on the viscosity of the gum, as shown in Figure S6. However, as the temperature increased, the effect of the NaCl was minimized, and the viscosity appeared almost to be the same as if no NaCl was added, as shown in Figures S7 and S8. For example, at 50 °C and a 0.01 s-1 shear rate, the gum viscosity was 48.34 mPa s; the viscosity of the gum with the NaCl was 36.12 mPa s. The Newtonian viscosity region was observed at a low shear rate (i.e., 0.04 s-1). Moreover, as the temperature increased, the effect of the salt solution decreased until it was almost negligible at 80 °C, as shown in Figures S9 and S10.

The temperature was kept constant to study the effect of adding PVA to the viscosity of XG at varying shear rates. It was found that the blending effect of PVA in the XG solution was minimum because the viscosity of the PVA and XG blend did not increase significantly compared to the viscosity of XG. For example, at 50 °C and a 1000 s-1 shear rate, the gum viscosity was 0.018 mPa s, while for the gum with PVA added, the viscosity was 0.020 mPa s as shown in Figure S9. In addition, at 80 °C and a 0.01 s-1 shear rate, the gum’s viscosity was 10.41 mPa s, while the gum with PVA had a viscosity of 11.37 mPa s as shown in Figure S10. However, the change in viscosity was minimal and considered negligible.

The viscosity of the XG was inversely proportional to the temperature and shear rate. Moreover, the experimental results from the effect of NaCl on the viscosity of the solution to simulate the injected water show that, the addition of 3% salt solution slightly decreased the viscosity of the xanthan solution. Furthermore, the results show that adding PVA barely increased the polymer’s viscosity, to the point that the change was negligible.

GA has a carboxyl group in its chemical structure, which was the reason it was chosen. It was expected to connect to the xanthan’s chain and increase the strength of the bonds among the molecules, and as a result, the viscosity would increase. This might happen with a chemical reaction. However, blending the xanthan gum with gluconic acid at the same concentration ratio (i.e., 1:1) slightly decreased the viscosity of the gum, as highlighted in Figure S11. The concentration used in the experiment was 1500 ppm (0.15%).

It was observed that mixing XG with AA did not increase the viscosity of the solution. It decreased the viscosity as the concentration of AA increased. The xanthan concentration used in the experiments was 1500 ppm and the temperature was 25 °C. The ratios of xanthan to AA used in the experiments were 1:1, 1:3, and 1:4, as shown in Figures S12 and S13. As the concentration of AA was increased in the blending process, the viscosity of the solution decreased. This reduction in viscosity resulting from adding the AA resulted from the reaction between the xanthan and AA. Therefore, the mixed solution became more diluted (1:4) and thus the viscosity decreased. A similar trend was observed at 50 °C as shown in Figure S14.

3.2.2. Chemically Modified Xanthan Gum—Xanthan Acrylate

The xanthan was modified with acrylic acid in a chemical reaction, as earlier described. The chemically modified xanthan gum is called Xanthan Acrylate and the analysis results are discussed below.

The viscosities of the Xanthan Acrylate and xanthan gum were measured at 25 °C and 50 °C and the concentration of both polymers was kept constant at 1500 ppm. The viscosity of both at 25 °C and 50 °C are shown in Figures S15 and S16. The Xanthan Acrylate showed a higher viscosity value than did the xanthan at the same shear rate. This improvement in viscosity was due to a chemical reaction with the acrylic acid that connected the hydroxyl group to the xanthan’s chain and made them stronger. Therefore, the viscosity increased.

The modified polymer offered higher stability, in terms of viscosity, with the temperature change, as shown in Figure 5. For example, at a shear rate of 0.1 s-1, the viscosities of the xanthan gum were 578 and 205 mPa s at 25 °C and 50 °C, respectively. The Xanthan Acrylate viscosities were 2200 and 1070 mPa s at 25 °C and 50 °C, respectively. The reduction in xanthan viscosity was around 65%. For the Xanthan Acrylate, the decrease in viscosity was around 50%. Even at a 10 s-1 shear rate, the xanthan gum viscosities were 63.5 and 44 mPa s at 25 °C and 50 °C, respectively. For the Xanthan Acrylate, they were 182 and 130 mPa s at 25 °C and 50 °C, respectively. The decrease in xanthan viscosity was around 30%. For the Xanthan Acrylate, the decrease in viscosity was around 20%.

Figure 5. Xanthan Acrylate viscosity vs. shear rate at 25 °C and 50 °C.

It is evident from the rheological experiments of the physical and chemical modifications that only the latter succeeded in increasing the xanthan’s viscosity. Consequently, a comparison between the xanthan gum and Xanthan Acrylate was conducted via core flood experiments to investigate which two offered more significant recovery.

COMMENT 12

Figure 6: Remove grid line from figure 6, 7, 8

Response:

Grid lines are removed from all the Figures.

According to Reviewer 2 suggestion, Figures 6 (recovery plot for pure XG) and 7 (recovery plot for modified XG) are combined to Figure 7 and Figure 6 and 7 are deleted from the manuscript.

COMMENT 13

Section 3.3: Add more discussion points and compare your results with other reports.

Response:

Section 3.3 is methodically revised, and more comparisons and discussions are added. Lines 327 to 366

3.3. Core Flooding Results

Core flooding simulates reservoir conditions and therefore helps investigate the EOR performances of polymer systems. Two core flood experiments were conducted to analyze and compare the recoveries of the pure xanthan and modified xanthan gums. The following sections present the recoveries of xanthan acrylate and pure xanthan gum and compare EOR performance.

3.3.1. Oil Recovery of the Xanthan Acrylate

First, the core was saturated with oil and 3% NaCl brine and then flooded by three pore volumes (PV) of 3% NaCl until no oil was produced. Then the core was flushed with Xanthan Acrylate with 3 PV until no more oil was produced. After that, the core was flooded by 1 PV of 3% NaCl brine to ensure all the mobile oil was produced.

The recovery obtained via water flooding was 43.85% of the initial oil in place (IOIP), and the tertiary oil recovery due to the Xanthan Acrylate flood was 19.21% of the IOIP. The total oil recovery by the water and xanthan acrylate was 63.1% of the IOIP. Figure 6 shows the recoveries in percentages, and Table 1 summarizes the core flood results by the Xanthan Acrylate solution.

Figure 6. Total oil recovery from the sandstone core after brine (3% NaCl) and 1500 mg/l ppm Xanthan gum acrylate mixture.

3.2.2 Comparison of Pure Xanthan Gum and Xanthan Acrylate Recoveries

The oil recovery performance of the modified xanthan gum was compared with the pure xanthan gum. A core flooding experiment was conducted using 3% NaCl and 1500 ppm pure xanthan gum solution. It was found that at the end of the waterflood, the oil recovery was 44.88%, and at the end of the pure xanthan gum solution flood, the tertiary oil recovery was 13.93%. The total recovery of this formulation was 58.81% of IOIP. Figure 7 and S17 demonstrates the recovery performance of pure xanthan and xanthan acrylate solutions. It is found from the literature that oil recovery by HPAM is usually around 12% to 17% [36].

The core flood results for the two formulations are given in Table 1 and Figure 7. It is clear from Figure 7 that the xanthan acrylate and 3% NaCl blend produced 19.21% tertiary oil. However, the pure xanthan and 3% NaCl solution recovered 13.93% tertiary oil, which was 5.28% less than the xanthan acrylate solution. Therefore, by comparing the two formulations, it can be concluded that the xanthan acrylate and 3% NaCl formulation were more efficient at recovering tertiary oil from the sandstone core.

Figure 7. Total oil recovery observed from the two formulations.

Table 1. Comparison of core flood experiments.

Core flooding

Xanthan

Xanthan Acrylate  

Polymer concentration (ppm)

1,500

1,500

Core length (inches)

6.04

5.94

Porosity

19.58

19.78

Permeability (md)

102.66

107.30

Swi %

39.88

40.61

Pore volume (cc)

34.00

33.83

Initial oil saturation %

60.22

59.49

No. of pore volumes injected

3

3

Initial oil volume (cc)

20.5

20.25

Residual oil after water flooding (cc)

11.3

11.37

Recovery % of IOIP after water flooding

44.88

43.85

Recovery % of IOIP after chemical flooding*

13.93

19.21

Recovery % of ROS by chemical flooding

25.27

35.18

                                                                                *Chemical flooding denotes, Xanthan and Xanthan Acrylate.

Round 2

Reviewer 1 Report

The manuscript can be accepted for publication.

Author Response

RESPONSE TO THE REVIEWER COMMENTS – Round 4

REVIEWER 1

Manuscript ID: polymers - 1373152

First, we would like to thank the reviewer for his/her favourable evaluation of our work and valuable suggestions. In the revised manuscript, we have addressed the specific comment made by the reviewer. In response to the reviewer comments, modified text in the manuscript is marked in red and heightened in yellow.

COMMENT 1

The average molecular weight of the modified xanthan gum should be compared with the pristine gum with GPC (gel permeation chromatography) method.

Response:

Please accept our sincere apology. Mohammed Said, the first author, was an international student. He had already graduated and left the country. It is difficult to provide the information at this stage. I tried, but no luck, sorry.

COMMENT 2

The manuscript should be checked for typo errors

Response:

Please accept our sincere apology. The manuscript is meticulously checked for typos and edited accordingly.
